# Mitochondrial Regulation of Ferroptosis in Cancer Therapy

**DOI:** 10.3390/ijms241210037

**Published:** 2023-06-12

**Authors:** Xiaoxia Cheng, Jiale Zhang, Yichen Xiao, Zhihang Wang, Jin He, Mengquan Ke, Sijie Liu, Qun Wang, Lei Zhang

**Affiliations:** 1School of Basic Medical Science, Henan University, Kaifeng 475004, China; 2School of Clinical Medicine, Henan University, Kaifeng 475004, China

**Keywords:** mitochondria, ferroptosis, cancer, iron, oxidative damage

## Abstract

Ferroptosis, characterized by glutamate overload, glutathione depletion, and cysteine/cystine deprivation during iron- and oxidative-damage-dependent cell death, is a particular mode of regulated cell death. It is expected to effectively treat cancer through its tumor-suppressor function, as mitochondria are the intracellular energy factory and a binding site of reactive oxygen species production, closely related to ferroptosis. This review summarizes relevant research on the mechanisms of ferroptosis, highlights mitochondria’s role in it, and collects and classifies the inducers of ferroptosis. A deeper understanding of the relationship between ferroptosis and mitochondrial function may provide new strategies for tumor treatment and drug development based on ferroptosis.

## 1. Introduction

Nowadays, cancer is still one of the deadliest diseases in the world. According to the current trend of significant cancer development, the cancer incidence rate worldwide will double by 2070 [1]. Finding new ways to treat cancer has become an urgent matter. In addition, widely used therapies such as radiotherapy, chemotherapy, gene therapy, and immunotherapy are gradually being commonly used for the clinical treatment of tumors. However, these treatment modalities have specific adverse reactions or are challenging to pinpoint precisely, and patients need to be treated multiple times [2]. Researchers are currently working on more effective and precise treatments, such as using gene-editing tools with CRISPR/Cas9 to correct DNA mutations in human T-cells to enhance the efficacy of cancer treatment [3].

Inducing tumor cell death is one of the effective pathways for tumor therapy. With the continuous deepening of cancer research, it has been found that various forms of cell death, such as apoptosis and ferroptosis, can induce the death of cancer cells. Conducting a profound study on the death mode of tumor cells will play an essential role in the pathological mechanism of tumor development and treatment. Among these, ferroptosis, a newly discovered iron-dependent method of regulated cell death (RCD), is associated with the occurrence and treatment response of various types of tumors, and the ferroptosis pathway associated with cancer also affects the growth and proliferation of cancer cells [4]. Conversely, ferroptotic injury can promote macrophage M2 polarization and trigger immunosuppression of tumor-associated inflammation, thereby favoring tumor growth [4].

In 1995, researchers discovered that extracellular cystine was identified as a nutrient for mammalian cells and found that deprivation of cystine can cause cell death, which manifests as glutathione(GSH) depletion and can be alleviated by lipophilic antioxidants α tocopherols or iron-chelating agent deferoxamine (DFO) [5,6,7]. In addition, overloading extracellular glutamate induces cytotoxicity in nerve cells by inhibiting cystine transportation, leading to GSH depletion and intracellular peroxide accumulation [7]. However, this cell death was considered oxidative poisoning [8]. These earlier studies established a preliminary understanding of glutamate overload, GSH depletion, and cysteine/cystine deprivation during iron and oxidative-dependent cell death, which are now considered essential features of ferroptosis.

In 2003, through high-throughput screening of synthetic lethal compounds, Stockwell et al. discovered a new anti-tumor compound called erastin that can trigger the lethal cytotoxicity of engineered tumor cells, leading it not to express their isogenic regular cell counterparts but the mutant Ras oncogene [9]. In 2008, a subsequent synthetic lethality screening assay study discovered two oncogenic-Ras selective lethal compounds, RSL3 and RSL5, that triggered the death of independent cells, resulting in apoptosis in BeJLR cells [10]. Cell death triggered by these two compounds differs from other dying methods and can be blocked by reactive oxygen species (ROS) scavengers and iron chelators. Dixon et al. studied the physiological mechanisms of this pattern of cell death in detail and named it ferroptosis [11].

In early research, one of the characteristics of ferroptosis is lipid peroxide (LPO) caused by increased ROS, and studies have shown that mitochondria are an important site of ROS production [12]. As an intracellular energy factory, mitochondria produce O^2−^/H_2_O_2_ and the energy required for biochemical reactions in vivo via difference in membrane potentials, involving NADH/NAD and QH2/Q as two isotopic groups. Moreover, it also produces various enzymes which can induce LPO in a pathological state [13]. In addition, mitochondria produce energy, ROS, and other components involving different metabolic pathways such as electron transport, oxidative phosphorylation, fatty acid β oxidation, tricarboxylic acid (TCA) cycling, iron metabolism, and calcium homeostasis to maintain normal body homeostasis. All these processes provide an energy base for cell metabolism. However, mitochondria also play a vital role in tumor cells, providing vast energy for proliferation [14].

In light of this, this review aimed to further understand the role of ferroptosis and mitochondria in cancer and drug research and development on treating cancer associated with ferroptosis.

## 2. Hallmarks of Ferroptosis

### 2.1. Morphological Hallmarks

Morphological features are cellular variation characteristics that can be directly observed by optical or electron microscopy and do not require the use of special stains for staining. Ferroptosis is one mode of RCD, but unlike other modes of RCD (Table 1), ferroptosis does not show similar morphological changes as observed in apoptosis. Still, necrotic characteristics such as plasma membrane rupture and cell swelling are often observed where ferroptosis occurs [15]. Yagoda et al. and Dixon et al. found that ferroptosis mainly cause mitochondrial membrane density increase, mitochondrial volume decrease, mitochondrial crest disappearance, and other mitochondrial-related morphological changes [11,16]. Although the ferroptosis inducers can cause oxidative DNA damage, the nucleus still maintains its morphology. In addition, Liu et al. found that autophagy promotes ferroptosis, so autophagy-related structures also appear in ferroptotic cells and tissues [17,18].

In addition, comparing ferroptosis with the newly discovered PANoptosis, it can be seen that they have some distinction in their morphological characteristics. PANoptosis is defined as an inflammatory PCD pathway activated by triggers such as RIPK1 and regulated by the PANoptosome complex, which possesses typical morphological features of apoptosis, necroptosis, and pyroptosis (Table 1) [19]. In contrast, the morphological characteristics of ferroptosis are mainly reflected in mitochondrial changes, which is enough to reflect its uniqueness.

### 2.2. Biochemical Hallmarks

#### 2.2.1. Iron Accumulation

Due to ferroptosis being an iron-dependent RCD, its biochemical characteristics are related to iron-accumulation-mediated biochemical events [20]. Dixon et al. found that iron chelators inhibit the occurrence of ferroptosis in vitro or in vivo [11]. Hou et al. observed that cellular labile iron concentration increases during ferroptosis induction [21]. Iron intake mediated by lactotransferrin (LTF), the transferrin receptor (TFRC), and transferrin (TF) can regulate ferroptosis by regulating ferroptosis sensitivity, which identifies certain molecular regulators related to iron homeostasis. However, whether different iron metabolism regulators play the same role in ferroptosis remains uncertain. More importantly, although multiple types of iron-dependent enzymes exist in other subcellular organelles, the process of enzyme coordination and its regulation of ferroptosis is unclear.

#### 2.2.2. Lipid Peroxidation

Lipid peroxidation is an important biochemical marker for ferroptosis. More particularly, polyunsaturated fatty acids (PUFAs) oxidation by ROS to produce lipid hydroperoxides (L-OOH) is the most vital characteristic of ferroptosis [22]. Researchers found that ROS play a cell-type-dependent role in initiating ferroptosis and are produced by the Fenton reaction, which is mitochondria- or nicotinamide adenine dinucleotide phosphate (NADPH) oxidase (NOX)-mediated [11,23,24]. The key enzymes of lipid peroxidation to initiate ferroptosis are the arachidonate lipoxygenase (ALOX) family, which includes ALOX5, ALOX3, ALOX12, ALOX15B, and ALOX15 in human cells [25,26]. In addition, various upstream lipid synthesis and metabolic pathways need to be activated to ensure the production of PUFAs for subsequent lipid peroxidation, in particular, acyl-Coenzyme A synthase long-chain family member 4 (ACSL4), which mediates the conversation between arachidonic acid (AA) and AA-CoA, thereby promoting lipid peroxidation and causing ferroptosis [27,28].

#### 2.2.3. Inhibition of Antioxidant Systems

Because LPO is the primary biochemical feature in ferroptosis, early discoveries of ferroptosis inducers such as erastin and RSL3 are associated with inhibiting antioxidant systems [11,29]. At present, researchers have confirmed three antioxidant systems, including the GSH, coenzyme Q10 (CoQ10), and tetrahydrobiopterin (BH4) pathways [11,30,31,32,33]. These systems can work separately or together to inhibit ferroptosis mediated by oxidative damage. The GSH system is the main pathway to limit ferroptosis. Erastin can inhibit the upstream regulator system Xc− (an amino acid antiporter) or the downstream effector glutathione peroxidase 4 (GPX4) of GSH from inducing ferroptosis [34]. In addition, erastin is also a potential activator of mitochondrial voltage-dependent anion channel 2/3 (VDAC2/3), highlighting the participation of mitochondrial dysfunction in erastin-induced ferroptosis [16]. However, in some cases, inhibition of GPX4 fails to cause ferroptosis, and the CoQ10 and BH4 systems have been observed to act in its place.

Doll et al. and Bersuker et al. found that in cells with defects in the GPX4 gene, apoptosis-inducing factor mitochondria-associated 2 (FSP1) was found to inhibit iron death as a new inhibitor. The N-terminus of FSP1 contains a myristoylation sequence that helps to position it on the plasma membrane [31,32]. CoQ10 plays a role in capturing lipophilic radicals and generating antioxidants on the plasma membrane [35]. Hence, FSP1 can modulate the non-mitochondrial CoQ10 antioxidant system on the plasma membrane to reduce oxide production and inhibit ferroptosis [31,32]. In contrast, ferroptosis occurs when the CoQ10 system is suppressed.

BH4 is a redox-active cofactor, with a specific antioxidant effect. Kraft et al. found that enzymes synthesizing BH4, such as GCH1, can eliminate lipid peroxidation and thus act as an almost complete inhibitor of ferroptosis [30].

### 2.3. Protein Concentration Changes

#### 2.3.1. GPX4

GPX4 uses GSH as a substrate to reduce L-OOH to lipid alcohols (L-OH), inhibiting erastin-induced ferroptosis. Yang et al. also found ferroptosis inducers such as erastin and RSL3, which inactivate GPX4 to inhibit tumor growth in xenograft mice [29]. However, GSH depletion and inactivation of GPX4 resulted in increased LPO, leading to ferroptosis [36].

#### 2.3.2. P53

The tumor suppressor p53 (TP53), through transcription and translation, produces P53, which is a cancer suppressor protein. In fibroblasts and specific cancer cells such as the human breast cancer cell line (MCF7) and human osteosarcoma cell line (U2OS), p53 inhibits the expression of SCL7A11. SCL7A11 promotes GSH synthesis by mediating cystine uptake and glutamate release, protects cells from oxidative stress, and prevents lipid peroxidation-induced ferroptosis [37]. When the concentration of P53 increases, it inhibits the biochemical effect of SCL7A11, which promotes the occurrence of ferroptosis and causes the death of cancer cells, thus playing a tumor-suppressive role. In addition, excess SCLS7A11 can inhibit P53-induced ferroptosis [38].

#### 2.3.3. ACSL4

ACSL4 is a member of the long-chain family of ACSL, which is involved in the synthesis regulation of arachidonic acid-CoA (AA-CoA) and adrenic acid-CoA (AdA-CoA). Doll et al. found that ACSL4 is essential in shaping the cellular lipidome and determining sensitivity resistance to ferroptosis [27]. It leads to lipid peroxidation by inhibiting GPX4. Loss of ACSL4 is accompanied by a decrease in AA- and AdA-containing PE species utilized as substrates to induce the ferroptosis cascade of events when GPX4 is inactive [27]. Therefore, when the concentration of ACSL4 increases, it leads to increased lipid peroxidation and causes the occurrence of ferroptosis [27].

At the same time, in addition to the above several proteins, some proteins affect the occurrence of ferroptosis (Table 2 and Table 3).

## 3. Mechanisms of Ferroptosis

Ferroptosis results from lipid peroxidation, which exceeds the range of cellular metabolism. Now we summarize the mechanisms of ferroptosis and focus on the mitochondrial regulation of ferroptosis in this review.

### 3.1. Mitochondrial Mechanisms of Ferroptosis

Cellular ROS are mainly produced by complexes I and III in the mitochondrial electron transfer chain (due to electronic leakage, complexes I and III have superoxide anion, which can promote lipid peroxidation and ferroptosis) [48,49]. 2-oxoacid dehydrogenase complexes (ODHc) can produce ROS at much higher rates than complex I, which has been shown in a number of papers [13]. It is composed of multiple copies of three enzyme components: oxoglutarate dehydrogenase (E1), dihydro-lipoamide succinyltransferase (E2), and dihydro-lipoamide dehydrogenase (E3) [13]. The ROS produced by ODHc during metabolism is mainly due to the reaction of its internal FAD/FADH2 (redox coenzyme) with oxygen molecules. FAD/FADH2 receive and release electrons during metabolism, which can combine with oxygen molecules to form highly reactive superoxide radicals (O^2−^), a type of ROS [50]. ROS production likely induces ferroptosis by accelerating the accumulation of lipid peroxidation. Mitochondrial GPX4 inhibits lipid peroxidation in mitochondria and diminishes ferroptosis (Figure 1).

Mitochondrial ferritin (FtMt) can stockpile iron and is located in cellular mitochondria, and has been demonstrated to be structurally and functionally similar to the cytosolic H-ferritin [51]. The expression of FtMt is confined to cellular mitochondria of the central nervous system and some other tissues with high oxygen consumption [52]. Erastin upregulates VDACs located in the mitochondrial membrane, which induces the occurrence of ferroptosis [11]. However, there is no change in VDAC2 and VDAC3 levels in erastin-treated FtMt-SY5Y cells, indicating that FtMt protects erastin-induced cells from ferroptosis in a specific way [11,53]. NOX is upregulated in erastin-treated cells, which could be vital in generating ROS in ferroptosis [11]. Furthermore, in the erastin-treated cells with FtMt overexpression, there is little increase in the NOX2 level, inferring that FtMt inhibits the production of ROS in ferroptosis [11]. As shown in Figure 1, erastin promotes ferroptosis by upregulating NOX2, VDAC2, and VDAC3 and inhibiting system Xc^−^ from synthesizing GSH [54].

### 3.2. GPX4 and Lipid Metabolism Mechanisms in Ferroptosis

GPX4 could alleviate cellular ferroptosis by providing mitochondria with GSH, a reductant that promotes the elimination of cellular ROS [55]. Extracellular cysteine could be transported into the cell via system Xc^−^ [11]. Cystine is converted to cysteine and binds with glutamate to form γ-glutamyl-cysteine (γGC) under glutamyl-cysteine ligase (GCL), a key enzyme in the process of GSH synthesis. After that, GSH synthase catalyzes the connection between γGC and glycine to form GSH [56]. GPX4 could convert GSH into GSH disulfide and transform noxious L-OOH into avirulent L-OH [57]. L-OOH also contributes to ferroptosis through another pathway. Free PUFAs and CoA are linked to form PUFA-CoAs under the action of ACSL4, and lysophosphatidylcholine acyltransferase 3 (LPCAT3) catalyzes the binding of AA/AdA-CoA and membrane PE to produce AA/AdA–PE. ALOXs promote the conversion of AA/AdA–PE to PE-AA-O-OH/PE-AdA-O-OH, thus resulting in ferroptosis [58]. As a tumor suppressor protein, p53 represses the expression of SLC7A11, a crucial component of system Xc^−^ (Figure 2).

Physiologically, PUFAs constantly undergo redox to achieve equilibrium. When oxidation processes surpass reduction processes, which exceed the cell’s capacity, this can result in plethoric lipid peroxidation and ferroptosis [11,59]. PUFAs can be oxidized under the function of multifarious lipoxygenases (LOX), cyclooxygenases (COX), and ACSL4, which promote lipid peroxidation (Figure 2).

In addition, CoQ10 is regarded as a lipophilic antioxidant that inhibits the propagation of LPO, and it constitutes the second mechanism in ferroptosis control. Intracellular NADPH production occurs through at least three pathways. NADP-dependent malic enzyme catalyzes the oxidative decarboxylation of malate to generate pyruvate, carbon dioxide and NADPH [60,61]. The pentose phosphate pathway is also a major source of NADPH [62]. Isocitrate is converted to α-ketoglutarate (α-KG) under dehydrogenase 1 (IDH1) and IDH2 to produce NADPH, while NADPH is used to produce CoQ10 [63]. Ferroptosis suppressor protein 1 (FSP1) promotes the transformation of the oxidation state of CoQ10 into the reduction state of CoQ10 to reduce lipid radicals (Figure 2) [31].

**Figure 2 ijms-24-10037-f002:**
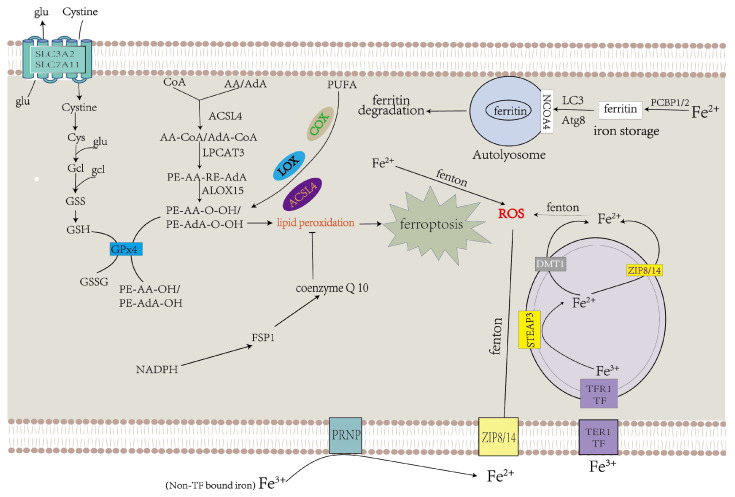
Abnormalities in iron and lipid metabolism induce ferroptosis. Insufficiently controlled intracellular or degraded iron storage and PUFA-enriched phospholipids are the premises for ferroptosis leading to cell death. System Xc^−^ transfers cystine into the cytoplasm, which is converted to cysteine and used to produce GSH, a necessary ingredient of GPX4 for eliminating LPO. CoQ10 also decelerates LPO [64].

### 3.3. Iron Metabolism in Ferroptosis

Under physiological circumstances, ferric ion (Fe^3+^) is transported into the cell under transferrin receptor 1 (TFR1) (Table 4), which takes up transferrin-bound iron (TBI) into cells by receptor-mediated endocytosis and then transforms into ferrous iron (Fe^2+^) via six-transmembrane epithelial antigen of the prostate 3 (STEAP3) in the endosome [65,66]. Compared with Fe^3+^, Fe^2+^ is more toxic to cells. Ferrous iron is transferred from the endosome to the cytoplasm via DMT1 (Table 4) [67]. ZIP8 and ZIP14 can transport non-transferrin-bound iron (NTBI) across the cell membrane (Table 4), and this process is intensified by ferrireductase in the prion protein PRNP which reduces Fe^3+^ to Fe^2+^ as one of its multiple functions [68,69,70,71,72,73]. Superfluous Fe^2+^ results in the accumulation of ROS by the Fenton reaction, which facilitates the occurrence of ferroptosis (Figure 2) [74].

As iron chaperones, Poly(rC)-binding proteins PCBP1 and PCBP2 bind to iron and transport it to ferritin for stockpile [75]. The iron-storage protein includes FTL and FTH1 (Table 4), which are degraded to release iron under the effect of lysosomes. LC3 has been best surveyed and regarded as an autophagosome marker in cells, constituting an Atg8-like conjugation system, named as the LC3-conjugation system [76]. Nuclear receptor coactivator 4 (NCOA4) mediates ferritinophagy, a kind of selective autophagy that catalyzes ferritin degradation, thus increasing free iron levels and promoting ferroptosis. The ubiquitin–proteasome system is engaged in the degradation (Figure 2 [77]).

**Table 4 ijms-24-10037-t004:** Protein associated with ferroptosis mechanisms.

Name	Function	Reference
TFR1	A membrane receptor that internalizes iron-bound transferrin through receptor-mediated endocytosis	[78]
FtH	Primary iron-storage protein that degrades to release iron under the effect of lysosomes	[79]
ACSF2	Increases the content of intracellular lipids to promote ferroptosis	[27]
ZIP8/14	Divalent metal transporter that transports non-transferrin-bound iron across the cell membrane	[80]
DMT1	Involved in transport of iron (Fe^2+^) from endosomes to the cytoplasm	[81]

## 4. Mitochondrial Function in Ferroptosis

### 4.1. The Role of Mitochondria in Inducing Ferroptosis

#### 4.1.1. Mitochondrial ROS Production and Lipid Peroxidation in Ferroptosis

Mitochondria play a crucial role in the production of ROS, the specialized molecules that regulate cell signaling and function and performs several essential functions [48]. Carsten Culmsee et al. demonstrated that mitochondrial ROS are significantly increased in HT-22 and MEF cells when stimulated with erastin [82]. Ferroptosis is the result of lipid peroxidation of the cell membrane, which ensures the participation of ROS, and mitochondria play an essential role in this process [56]. O_2_^•−^, produced in mitochondria, is converted to hydroxyl radicals (^•^OH) through complex processes. This hydroxyl radical (^•^OH) later extracts bis-allylic hydrogen from PUFAs to form lipid radical which reacts with oxygen to form PUFA-OO• and ultimately PUFA hydroperoxides (PUFA-OOH) (Figure 3) [83,84]. At the same time, excess iron can damage cells due to ROS production, eventually leading to ferroptosis [85]. Therefore, the production of ROS in mitochondria may lead to lipid peroxidation on cell membranes and, ultimately, ferroptosis.

#### 4.1.2. The Mitochondrial TCA Cycle May Cause Ferroptosis

Glutamine can be used as a nitrogen source for the anabolism of various substances and a carbon source for TCA in mitochondria, and mitochondrial glutaminase GLS breaks down glutamine to produce glutamate. Glutamate is then converted by glutamic-oxaloacetic transaminase (GOT1) to α-KG, fueling the TCA recycle process [86]. Studies have shown that inhibiting glutaminolysis can lead to ferroptosis [23,87]. The TCA cycle may regulate ferroptosis by supporting electron transport chains (ETC) that promote ROS production. Another possible route is that ETCs drive proton motive force, and ATP synthesis which inhibits AMP-activated protein kinase (AMPK), the enzyme which can block ferroptosis by inhibiting the synthesis of malonyl-CoA (a precursor for PUFA synthesis) [88].

#### 4.1.3. Iron Overload in Mitochondria Promotes Ferroptosis

Mitochondria are the centers of intracellular iron metabolism. The endosome-mitochondria “kiss and run” interaction is a significant pathway widely studied to transport iron into mitochondria [89,90]. Iron enters into the mitochondrial matrix through the outer mitochondrial membrane (OMM) and inner mitochondrial membrane (IMM). Iron is stored in FtMt in the mitochondrial stroma or used to synthesize heme and Fe-S clusters. These two iron-containing factors are likely associated with ferroptosis [51]. The mechanism by which iron is transported through OMM is still unknown, while transport on the IMM requires the assistance of the membrane transporter mitoferrin 1 (Mfrn1) and its homolog mitoferrin 2 (Mfrn2) [91]. Friedreich’s ataxia (FRDA) is caused by a decrease in mitochondrial proteins frataxin (FXN), and one of the features is an iron overload on the mitochondria [92]. At the same time, iron overload on mitochondria has been shown in various models of ferroptosis [93,94], but the underlying mechanism remains unclear. Iron overload on the mitochondria leads to mitochondrial dysfunction and lipid peroxidation in the mitochondrial membrane, leading to ferroptosis [95].

### 4.2. The Role of Mitochondria in Inhibiting Ferroptosis

There are three main types of cellular protective systems against ferroptosis. Each of these three protective systems has its own unique subcellular localization. GPX4 is localized to the cytoplasm and mitochondria, FSP1 is localized to the plasma membrane, and DHODH is localized to the mitochondria. Among them, the inhibition of ferroptosis by mitochondria is mainly through GPX4 and DHODH [96]. GPX4, a member of the glutathione peroxidase family, plays an important role in inhibiting ferroptosis. GPX4 is an inhibitor of lipid peroxidation, which can degrade small molecular peroxides and relatively complex lipid peroxides, and has shown inhibitory ability in adriamycin-induced lipid peroxidation [97]. As an enzyme in mitochondria, DHODH is involved in the electron transport process in the cellular respiratory chain. CoQH is a free-radical-trapping antioxidant that inhibits lipid peroxidation in the inner mitochondrial membrane. During CoQH production, DHO is oxidized to orotate via DHODH. DHODH is located on the outer surface of the IMM, and is then reduced to CoQH along with CoQ, which is produced mainly in mitochondria [98].

Mitochondrial DHODH and GPX4 are two major mitochondrial protective mechanisms against ferroptosis. Loss of one system makes the cell more dependent on the other, and loss of both defense systems leads to ferroptosis mainly induced by mitochondrial lipid peroxidation. In addition, GSH is abundant in mitochondria, and a variety of mitochondria-related proteins can mediate GSH entry into mitochondria to maintain mitochondrial REDOX homeostasis [99]. This further confirms the role of mitochondria in ferroptosis. Ferroptosis mediated by altered mitochondria function promotes tumorigenesis.

## 5. Ferroptosis Inducer

The burden of cancer incidence and death has been increasing rapidly worldwide in recent years. According to statistics, 19.3 million newly diagnosed cancers worldwide in 2020 were the leading cause of death in most countries/regions [100]. Traditional treatment methods, including surgery, radiotherapy, and chemotherapy, have advantages in tumor treatment but also have some limitations. Ferroptosis, as an adaptive mechanism to eliminate malignant cells, constitutes a new way of tumor inhibition. As we mentioned above, several molecules have been found to treat tumors based on the relevant mechanisms of ferroptosis.

Ferroptosis inducers can be divided into the following categories: (i) ferroptosis inducers targeting the production of ROS via iron accumulation (i.e., auriculasin and DHA); (ii) ferroptosis inducers targeting system Xc^−^ (i.e., erastin, sulfasalazine, bavachin, IMCA, and metformin); (iii) ferroptosis inducers targeting the consumption of GSH (i.e., sodium molybdate and Artesunate); (iv) Fferroptosis inducers targeting GPX4 (i.e., HNK, DET and DETD-35, RSL3); (v) ferroptosis inducers targeting DHODH (i.e., brequinar); and (vi) multiple targeted ferroptosis inducers (i.e., allomperatorin and bupivacaine) (Table 5).

### 5.1. Ferroptosis Inducers Targeting Different Mechanisms

#### 5.1.1. Ferroptosis Inducers Targeting the Production of ROS via Iron Accumulation

Iron accumulation in mitochondria produces mitochondrial ROS via the Fenton reaction, which uses ferrous ions as catalysts to convert H_2_O_2_ in cells into ROS to induce ferroptosis [101]. DHA accelerates ferritin degradation and then promotes ROS accumulation in cells to induce ferroptosis by regulating the activity of the AMPK/mTOR/p70S6k signaling pathway [102]. Auriculasin, isolated from Flemingia philippinensis, induces ferroptosis of colorectal cancer cells by increasing the accumulation of Fe^2+^ which increases ROS production in cells [103]. In conclusion, targeting the production of ROS via iron accumulation to induce ferroptosis may be an effective cancer therapy.

#### 5.1.2. Ferroptosis Inducers Targeting System Xc^−^

System Xc^−^, located in the cell membrane, mediates the transport of cysteine and glutamate. Glutamate is transferred to the outside of the cell, and simultaneously, cystine, which participates in the generation of GSH, is imported, thereby preventing the ferroptosis of cancer cells [104]. Therefore, some molecules inhibiting system Xc^−^ via several targets have been indicated for treating cancers by inducing cancer cell ferroptosis. Erastin, specifically inhibiting the system, has been demonstrated to be effective in treating a variety of tumors, including meningioma, diffuse large B-cell lymphoma (DLBCL), and renal cell carcinoma [29,105,106]. With the exact mechanism of ferroptosis induced by erastin, sulfasalazine, approved by the FDA, has been widely used in the treatment of endometrial cancer [107,108]. Bavachin induces ferroptosis in osteosarcoma cells, upregulating the expression of P53 and downregulating the expression of SLC7A11 via the STAT3/p53/SLC7A11 axis to achieve the purpose of treating osteosarcoma [109].

In the same way, metformin reduces the protein stability of SLC7A11 by inhibiting its acylation process from inducing ferroptosis in breast cancer cells [110]. Therefore, inhibiting system Xc^−^ may be feasible to cause ferroptosis of tumor cells and cure cancer.

#### 5.1.3. Ferroptosis Inducers Targeting the Consumption of GSH

GSH is an essential member of the cellular antioxidative system, loss of which will break redox homeostasis and cause ROS accumulation, eventually triggering the ferroptosis of cancer cells [111]. Artesunate, used as an anti-malarial drug previously, has been repurposed as an anticancer drug due to its induction of cell death in head and neck cancer via inducing ferroptosis by decreasing cellular GSH levels and increasing lipid ROS levels [112]. To sum up, reducing the levels of GSH may be feasible to induce ferroptosis in tumor cells.

#### 5.1.4. Ferroptosis Inducers Targeting GPX4

GPX4 converts L-OOH into non-toxic L-OH, preventing ferroptosis [32]. ML-162 induces ferroptosis in thyroid cancer cells by reducing the activity of GPX4 (Table 5 [113]. RSL3, a reagent containing electrophilic chloroacetamide, induces ferroptosis on DLBCL and renal cell carcinoma in vitro in a dose- and time-dependent manner through a decrease in the expression of GPX4 [29]. Therefore, the induction of ferroptosis by inhibiting GPX4 has been an effective therapeutic strategy for cancer.

#### 5.1.5. Ferroptosis Inducers Targeting DHODH

Dihydroorotate DHODH is reported to defend against ferroptosis, which operates dependent on mitochondrial GPX4 to inhibit ferroptosis in the IMM via reducing ubiquinone to ubiquinol, a radical-trapping antioxidant with anti-ferroptosis activity. In mitochondria, DHODH and GPX4 constitute two major ferroptosis defense systems by reducing the damage of LPO to mitochondria. Based on that, brequinar has been reported to damage the ferroptosis defense systems of low GPX4 tumors whose GPX4 defense system has low defense capability to inhibit their growth by inhibiting mitochondrial DHODH [96]. Therefore, inhibiting mitochondrial DHODH provides a new method to treat tumors.

#### 5.1.6. Multiple Targeted Ferroptosis Inducers

Some molecules induce ferroptosis through the joint action of the multiple targets above, including GPX4, SLC7A11, system Xc^−^, etc. Alloimperatorin induces breast cancer cell ferroptosis by reducing mRNA and protein expression levels of SLC7A11 and GPX4, which promotes the accumulation of Fe^2+^, ROS, and MDA, thereby inhibiting cell growth and invasion (Table 5) [114]. Bupivacaine, previously used as a local anesthetic, has recently been reported to modulate ferroptosis in bladder cancer by amplifying the level of Fe^2+^ and restraining the expression of system Xc^−^ and GPX4 [115]. Compared with inducers acting on a single target, acting on multiple targets to induce ferroptosis is hopeful for playing a more influential role in treating tumors.

### 5.2. Ferroptosis-Based Combinational Cancer Therapy

Combining some of the above molecules will have a better therapeutic effect. Chemical hybridization of sulfasalazine and DHA promotes the death of brain tumor cells [116]. Combined treatment with brequinar and sulfasalazine can induce ferroptosis, suppressing GPX4′s high tumor growth [96].

In addition, the combination therapy of some molecules and other substances also has a good effect on treating tumors. Combinational treatment of elesclomol and copper leads to consequent ferroptosis in CRC cells by promoting the degradation of SLC7A11, further enhancing oxidative stress [117].

In conclusion, combining two or more inducers has a better therapeutic effect on tumor treatment.

### 5.3. Nanomedicines Treat Tumors via Inducing Ferroptosis

With better targetability in inducing ferroptosis by recombination with other substances, new nanoparticles have been developed to treat tumors. IpGdIO-Dox, a novel magnetic nanocatalyst set through iRGD-PEG-ss-PEG-modified gadolinium engineering magnetic iron oxide-loaded Dox, is reported for MRI-guided chemo- and ferroptosis synergistic cancer therapies. Specifically, ipGdIO-Dox induces cancer ferroptosis by releasing abundant Fe^2+^ ions and then catalyzing H_2_LPCA into highly toxic OH•, which damages mitochondria and cell membranes (Table 5) [118]. Zero-valent-iron nanoparticle (ZVI-NP) enhanced GSK3 β/β-Trcp-dependent Nrf2 degradation by activating the AMPK/mTOR signaling pathway to selectively trigger ferroptosis in lung cancer cells (Table 5) [119]. Therefore, nanoparticles combined with other substances have better targetability of tumor cells.

**Table 5 ijms-24-10037-t005:** Inducers of ferroptosis.

Names	Targets	Mechanisms	Cancers	References
Erastin	System Xc^−^	Inhibition of the cystine–glutamate antiporter	Meningioma, Diffuse large B-cell lymphoma (DLBCL) and Renal cell carcinoma	[29,105,106]
Sulfasalazine	System Xc^−^	Inhibition of the cystine–glutamate antiporter	Endometrial cancer	[107,108]
Bavachin	STAT3/p53/SLC7A11 axis	Upregulating the expression of P53 and downregulating the expression of SLC7A11	Osteosarcoma	[109]
Metformin	SLC7A11	Inhibiting SLC7A11 acylation from reducing its stability	Breast cancer cells	[110]
Artesunate	GSH	Decreasing cellular GSH levels and increasing lipid ROS levels	Head and neck cancer	[112]
ML-162	GPX4	Reducing the activity of GPX4	Thyroid cancer	[113]
RSL3	GPX4	Decreasing the expression of GPX4	DLBCL and renal cell carcinoma	[29]
Brequinar	DHODH	Damaging the ferroptosis defense systems by inhibiting mitochondrial DHODH	GPX4low tumor	[96]
DHA	AMPK/ mTOR/p70S6k signaling pathway	Degradation of ferritin and accumulation of ROS	Acute myeloid leukemia (AML)	[102]
Auriculasin	Fe^2+^	Accumulation of Fe^2+^ to increase the production of ROS	Colorectal cancer	[103]
Alloimperatorin	SLC7A11 and GPX4	Accumulation of Fe^2+^, ROS, and MDA	Breast cancer cell	[114]
Bupivacaine	xCT and GPX4	Restraining the expression of xCT and GPX4	Bladder cancer	[115]
ipGdIO-Dox	Fe^2+^	Releasing abundant Fe(II) ions and then catalyzing H2O2 into highly toxic OH•		[118]
ZVI-NP	AMPK/mTOR signaling pathway	Enhanced GSK3 β/β-Trcp-dependent Nrf2 degradation	Lung cancer	[119]

## 6. Ferroptosis Inhibitor

As noted above, ferroptosis damage is associated with the development and progression of tumors. Many studies have revealed that excessive ferroptosis plays a critical role in the development of many brain diseases including Alzheimer’s disease, Parkinson’s disease, and Huntington’s disease; multiple cardiovascular diseases (CVDS) including ischemia/reperfusion (I/R) injury, heart failure (HF), and atherosclerosis; and various other diseases (such as ischemia/reperfusion injury during kidney transplantation and infectious diseases) [111,120,121,122,123,124].

As studies on ferroptosis continue to deepen, more and more inhibitors of ferroptosis have been reported. The application of inhibitors of ferroptosis can broaden the treatment of these diseases by blocking the ferroptosis pathways. Studies have shown that the mechanisms that inhibit ferroptosis include reducing iron content, inhibiting lipid peroxidation, increasing GSH levels, etc [125]. Several inhibitors of ferroptosis and their mechanisms of action are briefly discussed below.

Ciclopirox olamine (CPX) is reported to be an iron chelator which has broad-spectrum antifungal and antibacterial abilities to inhibit ferroptosis by eliminating excess iron ion [11]. Deferoxamine (DFO) is also a widely used iron chelator with therapeutic effects on traumatic spinal cord injuries [11].

Furthermore, by reducing lipid peroxidation and alleviating ferroptosis, puerarin (an isoflavone) can prevent HF caused by pressure overload in TAC-induced rat HF models [124].

Ferrostatin-1 (Fer-1) was identified as an ferroptosis inhibitor by Dixon et al. in 2012 [11]. Some recent studies have further suggested that cardiac I/R damage may trigger the accumulation of lipid peroxidation, such as phosphatidylcholine oxide, thereby inducing cardiomyocyte ferroptosis [126]. Fer-1 can reduce infarct area due to I/R damage by reducing lipid peroxidation, which shows long-term improvement in cardiac function. Bai et al. found that Fer-1 may reduce high-fat-diet-induced atherosclerotic lesions by reducing lipid peroxidation [127]. In addition, studies have shown that Fer-1 can increase GSH levels to inhibit ferroptosis in oligodendrocytes [128].

In summary, ferroptosis inhibitors play an important role in many of the above diseases, providing new ideas and new methods for the treatment of many diseases. However, 19.3 million newly diagnosed cancers worldwide in 2020 were the leading cause of death in most countries/regions, but the treatment of cancer is still a problem around the world and the existing treatment methods for various cancers are relatively unsatisfactory [100]. It is obvious that the research on ferroptosis inducers is more urgent than ferroptosis inhibitors. This is also why this review focuses on how to effectively treat cancer by inducing ferroptosis. Therefore, only ferroptosis inhibitors are briefly introduced here.

In addition, microRNAs and IncRNAs are also involved in the regulation of ferroptosis and play an important role. Table 6 summarizes some related microRNAs and IncRNAs, their mechanisms of actionm and clinical applications, which will not be repeated here.

## 7. Conclusions and Perspective

In conclusion, as a form of regulatory cell death, ferroptosis mainly depends on iron-mediated oxidative damage and subsequent cell membrane damage. The key to inducing ferroptosis is to increase iron accumulation, free radical production, fatty acid supply, and lipid peroxidation. In recent years, ferroptosis has shown exciting prospects in tumor treatment. A large number of known drugs and molecules have been proven to be able to treat ferroptosis. The treatment strategies of ferroptosis have also begun to diversify, which brings hope for tumor treatment. In the future, active areas will be exploring more signaling pathways of ferroptosis and new therapeutic approaches. Ferroptosis will play an increasingly important role in the treatment of tumors, and the development of new treatment methods will be a dynamic field.

## Figures and Tables

**Figure 1 ijms-24-10037-f001:**
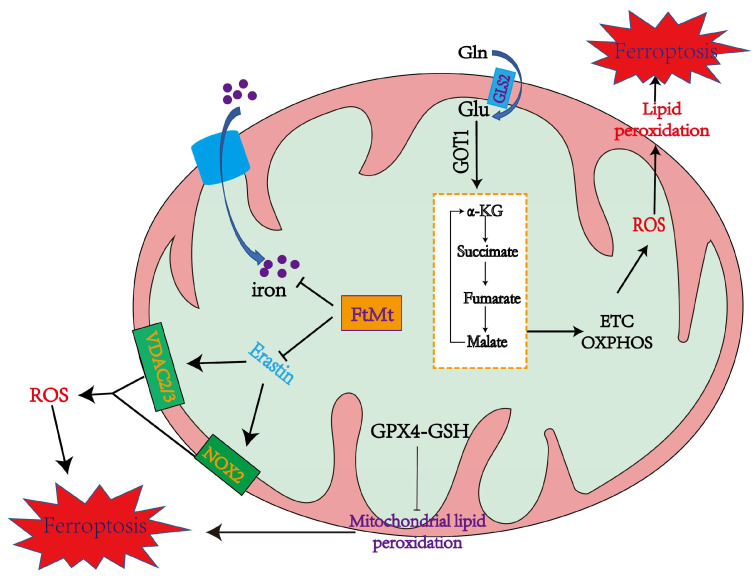
The mechanisms of mitochondria in promoting and inhibiting ferroptosis. Mitochondria coordinate necroptotic cell death through multiple mechanisms. The mitochondrial glutaminolysis-TCA-ETC axis is a significant pathway to induce ferroptosis, during which the production of ROS leads to ferroptosis through lipid peroxidation. GPX4 also inhibits lipid peroxidation in mitochondria. FtMt stores iron and inhibits Erastin from upregulating NOX2, VDAC2, and VDAC3 to alleviate ferroptosis [14].

**Figure 3 ijms-24-10037-f003:**
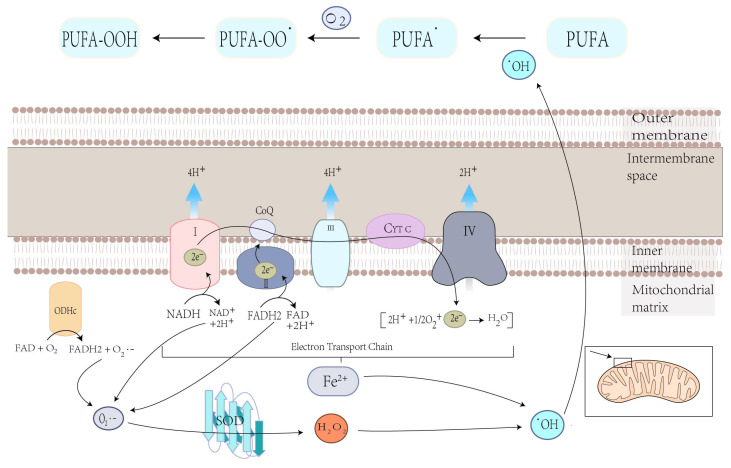
Electron leakage from the ETC cycle and the metabolic press of OGHDc produce O_2_^•−^, which is then converted to H_2_O_2_ under superoxide dismutase (SOD) mediation. H_2_O_2_ reacts with Fe^2+^ to produce hydroxyl radicals (^•^OH), which can extract the bis-allylic hydrogen in PUFAs from PUFA radicals (PUFA^•^). Then, PUFA^•^ reacts with oxygen from PUFA peroxyl radicals (PUFAOO^•^), which extracts hydrogen from another lipid to generate PUFA hydroperoxides (PUFA-OOH).

**Table 1 ijms-24-10037-t001:** Morphological hallmarks of cell death.

Type	Morphological Hallmarks
Ferroptosis	Mitochondrial membrane density increases, mitochondrial volume decreases, mitochondrial crest disappearance, the outer membrane of the mitochondria ruptures, and the nucleus is normal.
Apoptosis	Cells shrink, apoptotic bodies form, and surrounding tissues are normal.
Necrosis	The chromatin concentration changes, nucleus volume shrinkage, nuclear membrane rupture, and nuclear contours disappear.
Pyroptosis	Cells swell and cell membranes rupture, causing an inflammatory response.
Autophagy	A double-membrane autophagosome forms, cell membranes rupture, and organelles are engulfed.
PANoptosis	Cells swell, cell membranes rupture, and inflammasome formation occurs.

**Table 2 ijms-24-10037-t002:** Proteins downregulated in ferroptosis.

Protein	Function	Reference
FANCD2	Reduces GSH consumption and LPO production to inhibit iron accumulation and lipid peroxidation during ferroptosis through transcriptional and non-transcriptional-dependent mechanisms	[39]
NFE2L2	Encodes NRF2 and acts as an antioxidant and has anti-ferroptosis effects	[40]

**Table 3 ijms-24-10037-t003:** Proteins upregulated in ferroptosis.

Protein	Function	References
TFRC	Inputs iron into cells to promote ferroptosis	[41,42]
GLS2	It involves glutamine decomposition and promotes ferroptosis, regulated by P53	[43,44]
ATP5G3	Inhibition of its expression alleviates erastin-type ferroptosis	[45,46]
NCOA4	Participates in the control of ferritinophagy and free iron abundance and promotes ferroptosis	[47]

**Table 6 ijms-24-10037-t006:** The microRNAs and lncRNAs that regulate ferroptosis.

Names	Type	Mechanisms	Targets	Reference
miR-672-3p	microRNA, inducer	Differentially expressed after spinal cord injury (SCI) and induced ferroptosis by inhibiting FSP1	FSP1 and System Xc^−^	[129]
miR-129-5p	microRNA, inducer	Inhibits the expression of ACSL4	ACSL4	[130]
miR-5096	microRNA, inducer	Directly targets the SLC7A11 system and facilitates lipid peroxidation	SLC7A11	[131]
miR-541-3p	microRNA, inducer	Directly targets the GPX4 system and facilitates lipid peroxidation	GPX4	[131]
nuclear enriched transcript 1 (NEAT1)	incRNA, inducer	Inhibits the expression of ACSL4 and the occurrence of ferroptosis	ACSL4	[132]
miR-7-5p	microRNA, inhibitor	Indirectly reduces the labile iron pool and Fenton reactions	Transferrin	[133]
miR-522	microRNA, inhibitor	Reduces iron levels via another enzyme called arachidonate lipoxygenase 15 (ALOX15) to inhibit ferroptosis	ALOX15	[134]
miR-200a	microRNA, inhibitor	Targeting Keap1 and activating Nrf2 to inhibit ferroptosis	Keap1	[131]
RP11-89	IncRNA, inhibitor	Reducing iron level via regulating miR-129-5p to inhibit ferroptosis	miR-129-5p	[135]

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
