# Peer review of "Mitochondrial Regulation of Ferroptosis in Cancer Therapy"

_ijms, 2023, doi:10.3390/ijms241210037_

Round 1

Reviewer 1 Report

This review has a little in common with the previous review on ferroptosis from the same authors (https://www.mdpi.com/2218-273X/11/12/1790/htm), but the structure and idea of the current manyscript are different. Generally I found this review to be a good one, which still needs to be carefully reviewed.

Introduction is well written. As for the main part of the text, some chapters look too short. For example, chapter 2.3. It's information consisnts partially of the brief summary about several proteins, and a Table is given for a number of other proteins. The list of these proteins seems to have no system. As a result, it's one of the weakest chapters of this, otherwise good review.

In lines 304-305 it looks like this part of the text is not finished. Generally, the chapter 5 looks too short, so there is definetely something wrong here.

Generally, the chapter 6 is also a rather poorpy written one. From the logic of the review, it should have described a problem of ferroptosis in cancer. However, compared to the general mechanisms of ferroptosis described in first chapters, little attention is given to cancer. However, the description of inducers of ferroptosis, which could be used for cancer terapy is discussed extencively. 

Some other corrections and suggestions are provided below.

L. 171. "Cellular ROS are mainly produced by complexes I and III in the mitochondrial electron transfer chain..." Although this is a common point, the overall understading of ROS production currently agree, that there are other sources of ROS other than ETC. For example, 2-oxoacid dehydrogenase complexes can produce ROS at much higher rates than complex I, which was shown in a number of papers in recent decades. Please take it into account. This can be added to the Fig. 3 and corresponding text too.

When it comes to the complexes of 2-oxoacids such as OGDH, its function in regulation of redox metabolism, including cancer, may be be of real importance, and could contribute to this review.

L.222-225. You've forgotten about NADP-dependent malic enzymes.

L. 236. Several functions have ben proposed for PRNP, so a remark that the mentioned function is one of many is needed.

Chapter 6.1 - what are the letters (A), (B) and (C)? From these letters a figure is expected, but there are neither links to a figure nor such letters in Fig. 4.

Line 328 - a reference is needed (2000 kinds of cells).

The content of chapter 6.2 duplicates some of the previously provided data. Some content should be added, or the paragraph could be deleted.

Some sentences which caught my attention are listed below. These are only a few examples, but a carefull check is encouraged.

L. 23 - " in clinically treating tumors"

L. 36 - " inflammation-related inflammation immunosuppression"

The title of the paragraph 6 sounds a bit strange ("through").

A sentence in line 334.

Reviewer 2 Report

Cheng, X, Zhang, J., et al have discussed emphasizing the role of mitochondria based on the mechanistic insights of ferroptosis and categorize ferroptotic inducers. The authors considered that the mechanistic underpinning of ferroptosis and mitochondrial involvement in it has potential therapeutic applications in the diagnosis and treatment of tumors and drug development.

In this study, the authors demonstrated and discussed ferroptosis as a form of iron-dependent regulated cell death. The authors mentioned about the ongoing research on targeting ferroptosis has shown exciting prospects in tumor treatment and in the future exploring more signaling pathways of ferroptosis for tumor-targeted therapy would be a step forward.

The article provides a balanced view, the authors discussed several points with proper references related to ferroptosis, its induction, and mitochondrial crisis. One of the important parts is that all major statements were supported by the appropriate references relevant to this study.

But the novelty of this study had been discussed before by several other groups on “Regulatory pathways and drugs associated with ferroptosis in tumors”, “Mitochondria as multifaceted regulators of ferroptosis”, or “Ferroptosis and Cancer: Mitochondria Meet the “Iron Maiden” Cell Death” etc. I am struggling to understand what is new or the main significance of this study.

https://www.nature.com/articles/s41419-022-04927-1

https://academic.oup.com/lifemeta/article/1/2/134/6845830

https://www.ncbi.nlm.nih.gov/pmc/articles/PMC7349567/

https://pubmed.ncbi.nlm.nih.gov/34328510/

- Groups had already discussed several points similar or further to this study.

One of the interesting parts is that the authors also discussed the inducers of ferroptosis. All the charts containing, morphological hallmarks of cell deaths, proteins that affect ferroptosis, and inducers of ferroptosis are highly appreciated. Although one of the major and intriguing morphological hallmarks of cell deaths is missing –“PANoptosis”, which is highly relevant in discussing tumor biology.

https://www.sciencedirect.com/science/article/pii/S2001037021003287

Nonetheless, the article seemed to possess few major concerns. Overall, the clarity of the text is good and easily understandable but a few of the points need to be discussed properly with references and tables. The manuscript has very few typographical and grammatical errors which need to be corrected. In general, the manuscript can accomplish the caliber of quality for consideration for publication in the “International Journal of Molecular Sciences”. The authors are advised to consider the comments below:

Comments

1.      Ferroptosis is driven by several genes related to iron metabolism, lipid synthesis, and oxidative stress pathway. Table summarizing genes (with appropriate references) would be highly relevant to this study. Mentioning few important genes like transferrin receptor, TFR1; ferritin heavy chain, FtH; iron response element binding protein 2, IREB2, acyl-CoA synthetase family member 2, ACSF2; citrate synthase, CS etc

2.      A table summarizing microRNAs, and lncRNAs, acting as modulators of the ferroptosis process (with appropriate references) would be very relevant to this study.

3.      Discuss another morphological hallmark of cell death–“PANoptosis” in tumors.

4.      The application of inhibitors is also crucial in the study of ferroptosis. Please provide a discussion of why and how ferroptosis inducers have more significance than inhibitors. For guidance please follow this article: https://www.nature.com/articles/s41420-022-01297-7

5.      Figure 1 has a high similarity with a figure from a different manuscript. Either change the figure or mention it as a reference to the other paper.

https://www.mdpi.com/2075-1729/11/3/222

6.      Figure 2 – Redo the figure with better resolution and color markings. Use this article as a reference: https://link.springer.com/article/10.1007/s10495-022-01795-0

7.      Carefully check the author list organization *****

Overall, the clarity of the text is good and easily understandable. The manuscript has very few typographical and grammatical errors which need to be corrected.

Round 2

Reviewer 1 Report

The provided revision had practically fulfilled the questions or required corrections revealed during the first review. Still, a few small but important moments need to be clarified to improve the manuscript.

1) the phrase "2-oxoacid dehydrogenase complexes (OGDHc) produce ROS at much higher rates..." should contain the word "can" as in the cited paper [13]. In addition, it's strongly recommended to change the abbreviation. OGDH is a gene for 2-oxoglutarate dehydrogenase, which is one of the 2-oxoacid dehydrogenase complexes (see [13]). The abbreviation "OADH" can also cause misunderstanding... Choose ODHc or 2-OxDHc or something else instead, but not OGDHc or OADHc.

The problem is, it looks like you've cought yourself with this "OGDH", because already in the line 191 you cite OGDH as a single enzyme, while in lines 187-188 several ones are meant (see [13]). So, the paper [50] look not really on its place here. If you want discussing  2-oxoacid dehydrogenase complexes, it's better to mention them all and add a few references for each of them or cite a book or a good review on this topic then. That doesn't mean the discussion should be long.

To sum it up, the newly added paragraph has to be rewritten (maybe slightly shortened) to remove the described mistake.

In addition, these complexes added to the figure 3 (a wrong abbreviation is used) should be shown not as a membrane protein but as protein complex(es) localized in the mitochondrial matrix.

2) the correction about the malic enzyme is wrong. You've made a mistake with the place, probably because of a shift in line numbering. My remark was about the paragraph with the sentence "Intracellular NADPH production occurs through two crucial pathways." - In fact there are at least 3 pathways. You've forgotten about the NADP-dependent malic enzymes as another source of NADPH.

3) instead of "... ferrireductase in the prion protein PRNP which reduces Fe3+ to Fe2+ [66, 67]. PRNP function is only one of its functions in the cell [68]." I would suggest writing "... ferrireductase in the prion protein PRNP which reduces Fe3+ to Fe2+ [66, 67] as one of its multiple functions [...]." The reference [68] is fine, but some other references could be added, describing for example PRNP ability to bind thiamine or other facts. 

Although not many corrections remain to be done, I have to choose the option "major revision".

Sometimes used phrases sound not very common or easy to understand.

Formatting and typos should be checked. For example, the title "Combined treatment of cancer-based on ferroptosisnow likely has a typo.

Author Response

请参阅附件

Round 3

Reviewer 1 Report

Current version of the manuscript, provided by Xiaoxia Cheng et al. has fulfilled my previous questions and suggestions. I'd like only to point attention to the sentences "NADP-dependent malic enzyme catalyzes the oxidative decarboxylation of malate to generate pyruvate, carbon dioxide and NADPH [62, 63]. It is produced from the pentose phosphate pathway, essential for NADPH production."

Currently, the second sentence sounds strange and should be corrected, taking the context into account.

It's a pleasure to review your manuscript. I wish it could be more accurate from the very beginning. Good luck in experimental science too.

Language correction by a native speaker is recommended.
